# Injection of Ultra-Purified Stem Cells with Sodium Alginate Reduces Discogenic Pain in a Rat Model

**DOI:** 10.3390/cells12030505

**Published:** 2023-02-03

**Authors:** Hisataka Suzuki, Katsuro Ura, Daisuke Ukeba, Takashi Suyama, Norimasa Iwasaki, Masatoki Watanabe, Yumi Matsuzaki, Katsuhisa Yamada, Hideki Sudo

**Affiliations:** 1Department of Orthopedic Surgery, Faculty of Medicine and Graduate School of Medicine, Hokkaido University, N15W7, Sapporo 060-8638, Japan; 2PuREC/Bio-Venture, Shimane University, Izumo 693-8501, Japan; 3Japan Tissue Engineering Co., Ltd. (J-TEC), Gamagori 443-0022, Japan; 4Department of Advanced Medicine for Spine and Spinal Cord Disorders, Faculty of Medicine and Graduate School of Medicine, Hokkaido University, N15W7, Sapporo 060-8638, Japan

**Keywords:** low back pain, intervertebral disc regeneration, ultra-purified clonogenic bone marrow-derived mesenchymal stem cell, ultra-purified alginate

## Abstract

Intervertebral disc (IVD) degeneration is a major cause of low back pain. However, treatments directly approaching the etiology of IVD degeneration and discogenic pain are not yet established. We previously demonstrated that intradiscal implantation of cell-free bioresorbable ultra-purified alginate (UPAL) gel promotes tissue repair and reduces discogenic pain, and a combination of ultra-purified, Good Manufacturing Practice (GMP)-compliant, human bone marrow mesenchymal stem cells (rapidly expanding clones; RECs), and the UPAL gel increasingly enhanced IVD regeneration in animal models. This study investigated the therapeutic efficacy of injecting a mixture of REC and UPAL non-gelling solution for discogenic pain and IVD regeneration in a rat caudal nucleus pulposus punch model. REC and UPAL mixture and UPAL alone suppressed not only the expression of TNF-α, IL-6, and TrkA (*p* < 0.01, respectively), but also IVD degeneration and nociceptive behavior compared to punching alone (*p* < 0.01, respectively). Furthermore, REC and UPAL mixture suppressed these expression levels and nociceptive behavior compared to UPAL alone (*p* < 0.01, respectively). These results suggest that this minimally invasive treatment strategy with a single injection may be applied to treat discogenic pain and as a regenerative therapy.

## 1. Introduction

Low back pain (LBP) is a major health problem, with 70% of people experiencing LBP in their lifetime; severe LBP interferes with daily life [1,2,3,4]. In particular, chronic LBP is directly related to decreased quality of life and shortened life expectancy [2]. Intervertebral disc (IVD) degeneration is one of the main causes of LBP [5,6]. Treatment strategies for chronic LBP due to IVD degeneration include physical therapy, exercise, and medication, or surgery when conservative treatments are inadequate [5,6]. Physical therapy and medication improve LBP in approximately 20% of patients [6]. Spinal surgery has some efficacy in treating LBP; however, existing treatments do not directly approach the cause of IVD degeneration and/or discogenic pain [5,6]. No curative treatment for chronic LBP is established and new treatments to improve chronic LBP are needed.

The IVD comprises nucleus pulposus (NP) and annulus fibrosus (AF). The extracellular matrix (ECM) of the NP comprises glycosaminoglycans, proteoglycans, and type II collagen. It is highly hydrated, and the function of the NP is to disperse water pressure under compressive loading [7]. IVD degeneration is characterized by loss of hydration and ECM degradation, and this leads to changes in the overall biomechanics of the spine [7]. Furthermore, degenerated IVD tissue does not spontaneously regenerate in vivo due to poor nutrient supply and low cell division capacity [8]. Therefore, NP regenerative therapy is a highly attractive strategy to restore function in IVDs and is a potentially innovative LBP therapy [9,10,11,12].

Inflammation of the IVD is considered as the most important factor in acute LBP [13,14,15]. Chronic phase inflammation within the IVD and sensory nerve infiltration into the deep inner layers of the AF (called neoinnervation) are important factors known as discogenic pain [16]. Inflammatory mediators such as tumor necrosis factor (TNF)-α, interleukin-6 (IL-6), and nerve growth factor (NGF) are elevated in the IVDs of patients with LBP [17,18]. Moreover, TNF-α and IL-6 increase the production of NGF, and these changes play an important role in pain development [19,20,21]. Nerve growth factor exerts its effects via high-affinity tyrosine kinase A (TrkA) receptors and causes nerve ingrowth into the IVD [16,22]. Therefore, suppression of TNF-α, IL-6, TrkA, and neoinnervation in IVD could be a therapeutic target for discogenic pain.

Biomaterial-based or matrix-assisted cell based tissue regeneration therapies may be effective as a new treatment for IVD degeneration [10,11,18,23,24]. We previously reported that a non-cellular bioabsorbable ultra-pure alginate (UPAL) gel prevents IVD degeneration and reduces discogenic pain in a rat NP punch model [18]. Additionally, bone marrow-derived mesenchymal stem cells (BMSCs) are useful for patients with chronic LBP due to IVD degeneration [5,6,25]. A phase III trial shows that patients treated with chronic LBP due to IVD degeneration with allogeneic BMSCs had significantly less discogenic pain than those treated with saline solution or hyaluronic acid [25].

However, BMSCs used for clinical studies have a risk of variable (or even contradictory) findings since they include non-differentiating contaminant cells [26,27,28,29,30,31,32,33]. These issues are overcome by isolating cells, where using single-cell sorting is based on the expression of surface markers, and expanded to obtain genetically stable, ultra-pure, Good Manufacturing Practice (GMP)-compliant clonogenic BMSCs (REC: Rapid Expansion Clone) [27,28,29,30,31]. We demonstrated that implantation of RECs and UPAL gel combination further promotes IVD regeneration in the in vivo rabbit and sheep discectomy models, indicating that UPAL gel prevents cell leakage as a cell carrier and assists in BMSC activation [33,34].

This study hypothesized that administration of a REC and UPAL mixture reduces discogenic pain. We previously used UPAL as an intradiscal gel implant after discectomy in cases of lumbar IVD herniation. However, UPAL non-gelling solution can be administered into the IVD by a single injection without surgery in cases of chronic LBP without IVD herniation. This is expected to reduce discogenic pain and induce IVD regeneration by assisting in activating the remaining reparative NP cells and RECs as well as UPAL gel. In other words, intradiscal injection of REC and UPAL mixture within an outpatient clinic is expected to be a new treatment strategy for chronic LBP caused by IVD degeneration. The gelation procedure to prevent cell leakage is unnecessary, as there is no discectomy. This study evaluated the effects of injecting a mixture of REC and UPAL non-gelling solution on IVD tissue, inflammatory cytokines, and pain-related behaviors in a rat caudal NP punch model.

## 2. Materials and Methods

### 2.1. Animals

All animal procedures were approved by the Institutional Animal Care and Use Committee of Hokkaido University (17-0122) and performed in accordance with the ARRIVE guideline and National Institutes of Health guide for the care and use of Laboratory animals. Figure 1 provides an overview of this study. Outbred female Sprague–Dawley rats (12 weeks old, 260–300 g, n = 60) were obtained from Sankyo Labo Service Corporation (Tokyo, Japan). Sample sizes were the same as in the previous study, in which a minimum of 5 IVDs was assessed for each group [18]. In this study, we treated 2 IVDs per rat, ensuring 6 IVDs per group. Rats were adjusted for 7 days in cages at 23 ± 2 °C and 50 ± 10% humidity under standard laboratory conditions with a 12-h light/dark cycle as in the previously described experiments [18]. Three rats were kept in each cage and aseptically treated throughout the experiments; all efforts were made to minimize distress [18].

### 2.2. Injection of REC and UPAL Mixture in the Rat Caudal NP Punch Model

Immunohistochemical analysis, histological analysis, and behavioral nociception assays were performed using the rat caudal NP punch model according to previous studies [18,23] with modifications. Rats were randomly assigned to the following five groups: Intact control, sham (skin incision only), punch (NP punching only), UPAL (injection of UPAL solution after NP punching), and REC+UPAL (injection of mixed solution of REC and UPAL after NP punching). Each group had n = 3 rats and n = 6 IVDs. Two percent *w*/*v* UPAL solution was prepared by dissolving dry fibrous UPAL (400–600 mPa/s, Mochida Pharmaceutical Co., Ltd., Tokyo, Japan) in 0.9% (*w*/*v*) saline solution (Otsuka Pharmaceutical Co., Ltd., Tokyo, Japan) as previously described [18]. Frozen GMP-compliant RECs (Japan Tissue Engineering Co., Ltd., Aichi, Japan) were thawed in a warm bath at 37 °C and directly mixed with 2% (*w*/*v*) UPAL solution at a final cell concentration of 1 × 10^6^/mL [34].

A longitudinal incision at the coccygeal (Co) level was made in the dorsal tail skin and connective tissues under anesthesia with isoflurane (5% for induction, inhalation), followed by intraperitoneal administration of a mixture of 75 mg/kg ketamine and 0.5 mg/kg medetomidine [18]. An 18-gauge (G) needle was used to puncture through the AF to NPs of Co 4/5 and 5/6 (puncture diameter: 1 mm, depth: 2 mm) [18]. This was immediately followed by the injection of UPAL solution (4 μL per IVD) or REC/UPAL mixture (4 μL per IVD, final concentration 1 × 10^6^/mL) into the IVD from a different route than the NP punch. This was performed using a micro syringe with a 26G needle (Hamilton, Reno, NV, USA). No solution leaked from the punched hole during and after injection of the solution. The surgical site was irrigated with normal saline 5 min after injection, and the skin and connective tissue were closed with intermittent sutures. Co 4/5 and 5/6 were treated identically for each rat (the UPAL group received UPAL solution in both discs, the REC+UPAL group received a REC and UPAL mixture in both discs, while the punch group remained untreated after the NP punch in both discs).

### 2.3. Immunohistochemical Analysis

Immunohistochemical analysis was performed to detect TNF-α, IL-6, and TrkA levels on postoperative days 1, 7, and 28. Euthanasia was performed by cervical dislocation under deep anesthesia with isoflurane and operated rat tails (Co 4/5-5/6) were resected. Resected IVDs were fixed in 4% (*w*/*v*) paraformaldehyde for 48 h, demineralized in Christensen demineralizing solution for 1 week, washed in tap water for 24 h, and embedded in paraffin [18,23,24,35]. Transverse sections (5 µm thick) of the central part of the IVD were obtained [18]. Sections were deparaffinized with xylene, then incubated with proteinase K (Dako, Agilent Technologies, Santa Clara, CA, USA) at 37 °C for 15 min. Background peroxidase activity was blocked with 1% (*w*/*v*) hydrogen peroxide in methanol for 30 min at 37 °C, followed by incubation with 2% (*w*/*v*) bovine serum albumin for 30 min at room temperature (23 ± 2 °C). Specimens were incubated overnight at 4 °C with the following primary antibodies; anti-TNF-α mouse monoclonal antibody (1:50, Abcam, Tokyo, Japan, Cat# ab220210), anti-IL-6 mouse monoclonal antibody (1:500, Abcam, Cat# ab9324), and anti-TrkA mouse monoclonal antibody (1:100, Abcam, Cat# ab86474) [18,36]. Histofine Fast Red II (1:50, alkaline phosphatase substrate kit, Nichirei Bioscience Inc., Tokyo, Japan) was used for TNF-α analysis, HistoGreen substrate kit for peroxidase (1:25, substrate kit, Cosmo Bio Co., Ltd., Cat# E109, Tokyo, Japan) was used for IL-6, and Histofine DAB (1:25, substrate kit for peroxidase, Nichirei Bioscience Inc., Tokyo, Japan) was used for TrkA analysis [18,37]. Nuclei counterstaining was performed using hematoxylin for TNF-α or TrkA, or nuclear fast red for IL-6 [18].

One section per disc was evaluated and cells positive for TNF-α, IL-6, or TrkA were counted separately in 5 randomly selected independent fields using an optical microscope (40× magnification; Olympus, Tokyo, Japan) [11]. The number of positive cells for each stain was calculated as a percentage of the total number of cells in each of the NP or AF tissues. All image evaluations were performed by two independent blinded observers. Each observer performed three evaluations on one specimen, and quantitative data are presented as all individual means for each group.

### 2.4. Histological Analysis

Histological analysis was performed to assess postoperative IVD degeneration at 28 days. Rat tail IVDs (n = 3; n = 6 IVD per group) were harvested in the same manner as for IHC. Intervertebral discs were fixed in 4% (*w*/*v*) paraformaldehyde for 48 h, demineralized in Christensen’s demineralizing solution for 1 week, washed under running tap water for 24 h, and embedded in paraffin [20]. Midsagittal sections (5 µm thick) of the IVDs were obtained and stained with hematoxylin and eosin (H&E) to assess cell and tissue structure. Tissue structure and ECM were assessed using safranin O staining, while alcian blue (AB) staining was used to analyze ECM [18]. Histological scores were evaluated using a semiquantitative histological scoring system for IVD degeneration devised by Rutges et al. [36], as previously described [18]. This classification evaluates the anatomical structure of NP and AF using six categories: Endplate, AF morphology, AF-NP boundary, NP cellularity, NP matrix, and NP matrix staining, each scored from 0 (non-degenerative) to 2 (severely degenerative), and the sum of scores was calculated from 0 (healthy IVD) to 12 (completely degenerated IVD) [36]. Evaluation involved one section per disc, and all image evaluations were performed by two independent, blinded observers (20× magnification). Each observer made three assessments per specimen, and quantitative data are presented as the mean of all indices for each group.

### 2.5. Behavioral Nociception Assays

Thirty rats were sacrificed for IHC (n = 3 rats/group). Rats were subjected to Hargreaves, von Frey, and tail-flick tests as nociceptive behavioral tests, according to previous studies [18]. All behavioral tests were conducted by the same blinded observer. The rats were acclimated to the test environment for 20 min each, 24 h prior to, and immediately before the test [18]. The mean was calculated from multiple measurements for each rat (n = 6 rats per group) [18].

#### 2.5.1. Hargreaves Test

Hargreaves tests were conducted on preoperative day 2 (day—2) and postoperative days 2, 7, 14, and 27 using a Hargreaves device (Ugo Basile Biological Instruments, Gemonio, Italy) [18]. Rats were placed in individual spaces on a glass plate and exposed to infrared light (as a thermal stimulus) on the ventral surface of the tail opposite the skin incision; the retraction latency to the thermal stimulus was recorded [18]. Beam intensity was set at 50% of the maximum power, and the cutoff time was set at 20 s to prevent tissue damage [18]. Rats were given 4 trials on the same day, and rested for at least 1 min between trials.

#### 2.5.2. Von Frey Test

The von Frey test was carried out on preoperative day 2 (day—2) and postoperative days 2, 7, 14, and 27 using a dynamic plantar aesthesiometer (Ugo Basile Biological Instruments) [18]. Rats were placed in individual compartments surrounded by a wire net floor and a cap with air vents was stimulated with a filament (0.5 mm diameter, nickel-titanium alloy) on the ventral surface of the base of the tail (opposite side of the wound) [19]. The stimulus force was linearly applied from 0 to 5 g over 10 s, then held constant at 5 g for 30 s [18]. The time required to show nociceptive responses (flicking, licking, pulling, and shaking at the base of the tail) was defined as the tail-pulling latency; this was recorded as the sensory threshold [18,38]. Five tests were performed on each rat with at least a 10-s break in between tests.

#### 2.5.3. Tail-Flick Test

The tail-flick test was performed using a heat flux radiometer (Ugo Basile Biological Instruments) on preoperative day 1 (day—1) and postoperative days 3, 8, 15, and 28 to protect against tissue damage due to excessive thermal stimulation [18]. Each rat was acclimatized for 10 min by covering the body and limbs with a towel and placing only the tail on the device [18]. Infrared radiation was applied as a thermal stimulus to the ventral surface of the tail (5 cm from the distal end of the tail) to avoid tail response to the thermal stimulus, and the latency was recorded [18,39]. The cutoff time was set at 20 s to prevent tissue damage [18]. Rats were given 4 trials on the same day, and rested for at least 15 s between trials.

### 2.6. Statistical Analysis

The sample size of the quantitative data was decided with reference to previous reports [18,23,24]. All data are shown as the mean ± standard error (SE). One-way analysis of variance (ANOVA) and the Tukey-Kramer post hoc test were used for multiple-group comparisons, and an unpaired t-test was used for two-group comparisons. All statistical calculations were performed using JMP Pro-version 14.0 statistical software (SAS Institute, Cary, NC, USA) with a significance level of *p* < 0.05.

## 3. Results

The results of the intact control group were excluded in this section for clarity and are presented as Appendix A since the sham group and the intact control group had comparable results in all experiments, including immunohistochemistry (IHC), histological analysis, and behavioral nociception assays.

### 3.1. Injection of a REC and UPAL Mixture after IVD Punching Suppresses Inflammatory Cytokine Production

The percentage of TNF-α-positive NP and AF cells in the punch group presented a gradual increase from postoperative day 1 to day 28 (Figure 2a–d). The percentage of TNF-α-positive NP and AF cells in the UPAL group increased from postoperative day 1 to day 7, but decreased by day 28. However, the percentage of these positive cells was significantly lower than in the punch group at all the time points (*p* < 0.01 at each time point) (Figure 2b–d). The percentage of TNF-α-positive NP and AF cells was very low in the REC+UPAL group, and comparable to the sham group at each postoperative time point. Moreover, it was significantly lower than the punch group (punch vs. REC+UPAL, *p* < 0.01 at each postoperative time point), and significantly lower than the UPAL group on days 1 and 7 (UPAL vs. REC+UPAL, *p* < 0.01 on days 1 and 7) (Figure 2b–d).

The percentage of IL-6-positive NP and AF cells gradually increased in the punch group at each postoperative time point, while the percentage of IL-6-positive NP and AF cells in the UPAL group increased from day 1 to day 7, but decreased on day 28, and showed similar results to TNF-α (Figure 3a–d). The percentage of IL-6 positive NP and AF cells was very low at each postoperative time point in the REC+UPAL group. There was no significant difference compared with the sham group, but it was significantly lower than the punch group (punch vs. REC+UPAL, *p* < 0.01 at each time point). The percentage of IL-6 positive NP cells was significantly lower in the REC+UPAL group than in the UPAL group on postoperative day 7 (*p* < 0.01), and for IL-6 positive AF cells, the percentage was significantly lower on days 1 and 7 (*p* < 0.01, *p* < 0.01, respectively) (Figure 3b–d).

These results suggest that the REC and UPAL mixture inhibited the inflammatory response caused by IVD punching more than the UPAL solution alone.

### 3.2. Injection of a REC and UPAL Mixture Suppresses the Increase in TrkA Expression after IVD Punching

The percentage of TrkA-positive NP and AF cells in the punch group showed a gradual increase from postoperative day 1 to day 28 (Figure 4a–d). The percentage of TrkA-positive NP and AF cells in the UPAL group increased from postoperative day 1 to day 7, but decreased by day 28; the number of TrkA-positive cells was significantly lower than in the punch group (*p* < 0.01 at each time point) (Figure 4b–d). The percentage of TrkA-positive NP and AF cells was very low in the REC+UPAL group. There was no significant difference compared with the sham group at each postoperative time point (on each day). However, the values were significantly lower than the punch group (*p* < 0.01 on each day). The percentage of TrkA-positive NP and AF cells was significantly lower in the REC+UPAL group than in the UPAL group on postoperative days 1 and 7 (*p* < 0.01 on days 1 and 7) (Figure 4b–d).

These results suggest that the REC and UPAL mixture suppressed intradiscal TrkA receptor expression caused by IVD punching more than the UPAL solution alone.

### 3.3. Injection of REC and UPAL Mixture Inhibits IVD Degeneration

We previously reported that implantation of UPAL gel alone or UPAL gel in addition to BMSCs prevents IVD degeneration in various animal models of NP punch or partial discectomy [18,24,33,34]. However, the effect of non-gelatinized UPAL solution or a mixed REC and UPAL solution preventing IVD degeneration is unconfirmed. This study histologically examined their effect on IVD degeneration by injecting them into the rat caudal punched IVDs.

The endplates of IVD were regularly aligned in the sham group. Furthermore, AF tissue was well organized, the boundary between AF and NP was clear, there was no NP cell aggregation, and the ECM of NP tissue was well organized (Figure 5a). Meanwhile, the IVDs of the punch group had severe irregularities of the endplates, there was disruption of the annular structure, obscuration of the boundary between AF and NP, accumulation of NP cells, complete collapse and disappearance of the NP matrix, and cartilage nests of NP tissue (Figure 5a). The UPAL group had regular endplates, persistence of semi-cyclic structures with tortuous lamellar, mixed populations of NP cells, and some disruptions of the NP matrix structure (Figure 5a). Meanwhile, the IVDs in the REC+UPAL group maintained a semi-cyclic structure, including regular endplates, slightly tortuous but nearly normal lamellae, scattered NP cells with nearly no clusters, and NP matrix structure (Figure 5a). The scores for each of the categories are presented in Appendix A. AB staining indicated that IVDs in the punch group showed slight staining of the NP matrix, while those in the sham group indicated strong staining. The UPAL and REC+UPAL groups had similar or slightly reduced staining than the sham group (Figure 5a, column AB). The semi-quantitative histological IVD degeneration scores [37] were significantly lower in the REC+UPAL and UPAL groups on postoperative day 28 than in the punch group (*p* < 0.01, *p* < 0.01, respectively), with no significant difference among the sham, UPAL, and REC+UPAL groups (Figure 5b).

These results suggest that intradiscal injection of UPAL solution or a REC and UPAL mixture prevents IVD degeneration after NP punching.

### 3.4. Injection of REC and UPAL Mixture Reduces Nociceptive Behavior

Immunohistochemistry and histological analysis indicated that intradiscal administration of REC and UPAL mixture inhibits the expression of inflammatory cytokines and TrkA, and prevents degeneration of IVD tissues. This indicated the possibility of reducing pain due to IVD punching.

The Hargreaves test showed that the punch group had significantly shorter withdrawal latency than the other groups at each postoperative point, except for day 2 in the UPAL group (punch vs. sham or REC+UPAL, *p* < 0.01 at each time point, punch vs. UPAL, *p* < 0.01 on days 7 and 27) (Figure 6a). The UPAL group had shorter latency on postoperative days 2 and 14 than the REC+UPAL group (UPAL vs. REC+UPAL, *p* < 0.01 on days 2 and 14) (Figure 6a).

The von Frey test displayed that postoperative withdrawal latency significantly decreased in the punch group compared to the other groups (punch vs. sham, UPAL, or REC+UPAL, *p* < 0.01 at each postoperative time point). Postoperative latency in the sham group, UPAL group, and REC+UPAL group was equivalent after 7 days (Figure 6b).

Postoperative tail-flick test latency in the punch group increased after 8 days and was significantly higher than the other groups (punch vs. sham, UPAL, or REC+UPAL, *p* < 0.01 at days 8, 15, and 28), while the latency time in the sham, UPAL, and REC+UPAL groups remained unchanged until postoperative day 28 (no significant difference between groups at each time point) (Figure 6c).

These results suggest that the REC and UPAL mixture reduces pain caused by IVD punching and its effect is greater than the UPAL solution alone.

## 4. Discussion

Injection of REC and UPAL solution suppressed the production of TNF-α, IL-6 and TrkA, and inhibited IVD degeneration. Moreover, REC and UPAL injection reduced nociceptive behaviors. These results indicate that the REC and UPAL mixture may be a treatment strategy for LBP owing to the prevention of IVD degeneration, reduction of inflammation, and inhibition of neoinnervation based on the decreased expression of TNF-α, IL-6, and TrkA.

Inflammatory cytokines, such as TNF-α and IL-6, are increased in painful IVDs [17,21,23,40,41]. Furthermore, TrkA binds to NGF produced by IVD inflammation resulting in neoinnervation in the IVD and discogenic pain [16,22,40]. Implantation of hyaluronic acid gel or UPAL gel in rat IVD punch models suppresses pain-related behaviors by reducing the expression of inflammatory cytokines (TNF-α and IL-6) and NGF receptor (TrkA) [18,23], and decreased TrkA expression is associated with suppression or delayed neoinnervation in the IVDs [18]. This study showed that the injection of REC and UPAL mixture of non-gelling solution suppressed the expression of TNF-α, IL-6, and TrkA, and improved pain-related behavior. This suggested that REC and UPAL solution suppresses discogenic pain by the same mechanism as previously reported: Suppressing nociception, hyperinnervation, and nociceptive marker expression via the attenuation of key inflammatory signaling molecules and the modulation of protein regulatory pathways [18,23].

Bone marrow-derived mesenchymal stem cells suppress inflammation by inducing the production of anti-inflammatory cytokines [42,43]. Intra-articular autologous injection of BMSCs into the knee joint of patients with knee osteoarthritis reduces serum TNF-α and IL-6 levels and significantly improves pain and clinical scores compared with endoscopic synovectomy and hyaluronic acid injection [43]. This study showed that the REC+UPAL group further suppressed the production of inflammatory cytokines, TrkA, and pain-related behaviors compared with the UPAL alone group. These results indicated that adding REC to the UPAL solution exerted an anti-inflammatory effect induced by BMSCs shortly after administration, and reduced pain compared to UPAL alone.

The tissue regenerative abilities of UPAL and REC for IVD are demonstrated in various animal IVD discectomy models [18,24,33,34]. Cell-free UPAL gel implantation inhibited IVD degeneration in rat, rabbit, and sheep models of discectomy or punching [18,24]. This indicates that the accumulation and increase in remaining NP progenitor cells in UPAL gel promotes spontaneous regeneration of IVD tissue. Additionally, implantation of a mixture of UPAL gel and BMSCs/RECs was more effective in inhibiting tissue degeneration compared with cell-free UPAL gel alone in a rabbit and sheep model [33,34]. Coexistence of RECs and NP cells leads to the production of ECM and growth factor and causes RECs to differentiate into NP cells which results in IVD regeneration by BMSCs/RECs [33]. The gel encapsulation does not represent a mechanism of IVD regeneration, but rather the technique is used to prevent cell leakage and assist in BMSCs/RECs activation [33]. Similarly, the key role of non-gelling UPAL solution presented in this study was a cell carrier. Recent phase III trial showed that a single injection of allogeneic mesenchymal precursor cell with hyaluronic acid carrier significantly reduced chronic LBP, while hyaluronic acid alone was no different than saline [25].

Regenerative medicine using stem cells for lumbar degenerative diseases has attracted attention in recent years. New treatments using BMSCs for discogenic chronic LBP were developed and are undergoing clinical trials [5,6,25]. Local administration of autologous/allogeneic BMSCs to IVDs significantly relieves pain in patients with chronic LBP compared with the placebo [5,25]. However, these BMSCs have several limitations including high patient invasion, infection risk, cost for autologous BMSCs [6,26], and non-differentiating contaminant cells for allogeneic BMSCs [27,28,29,30,31,32,33]. Safer and higher quality BMSCs are hypothesized to be important for clinical applications. Rapidly expanding clones are ultra-purified stem cells that are distinguished from conventional BMSCs through superior cell proliferative capacity, cell size uniformity, and cell surface marker expression, i.e., the percentage of CD90-positive cells were 99.6% and CD45-positive cells were 0.0% [27,28,29,30,31,33]. In addition, the previous study demonstrates that the expression levels of NP cell markers, growth factors, and ECM components were significantly increased in the three-dimensional (D) co-culture of human NP cells and RECs compared with those observed in the 3D co-culture of NP cells with commercially available BMSCs [33]. Furthermore, tumorigenesis was previously analyzed in histological sheep IVD specimens during the 24-week evaluation period, suggesting that the combination of RECs with the gel did not induce tumorigenesis [33]. This study showed that the injection of mixed UPAL non-gelling solution with RECs into the IVD suppresses pain and tissue degeneration inhibitory effects. Injection of REC and UPAL mixture into IVDs is expected to be a clinically valid treatment strategy of discogenic chronic LBP.

There are several limitations to this study. First, it was not possible to evaluate the IVDs that cause discogenic pain due to tissue degeneration since the rat caudal IVD punch model was used. Second, evaluating pain-related behavior in the caudal vertebrae of rats cannot be considered equivalent to the evaluation of discogenic pain in the human lumbar spine where load bearing is applied. The lumbar disc injury model was used to evaluate disc pain in rat models [44,45,46]. This study used the caudal disc injury model since there are no significant differences in measuring pain between the lumbar and caudal vertebrae in rats. Third, this study was conducted only up to day 28 after injection of REC and UPAL solution into IVDs, and no further long-term follow-up was conducted. However, 28 days in SD rats is equivalent to several years in humans [47]. Therefore, this observation period was deemed to be sufficient to investigate treatment effects on chronic LBP in rats. Fourth, NP cell markers, growth factors, and ECM components expression were not determined in the IVD specimens since we previously showed expression profiles of these markers in human NP cells and REC in vitro [33] and rabbit NP and BMSCs in vivo [34]. Although the present study focused on behavioral changes related to IVD pain, these results might be the stronger evidence to support the IVD regeneration effect by the studied methods. Fifth, three rats per group may be insufficient to support biological replication, and an increase to 5 or 6 rats per group is recommended. Finally, magnetic resonance imaging (MRI) or micro-computed tomography (CT) assessments were not performed in this study. Although 1.5 tesla (T) and 3.0T MRI are widely used to evaluate human IVD degeneration, these magnetic strengths are not precise for IVD imaging in small animals [48]. We previously used both 7.0T MRI and micro-CT to assess murine IVDs [48]. However, these modalities are currently unavailable at our facility [48].

## 5. Conclusions

A novel treatment method was developed by injecting a mixture of REC and UPAL non-gelling solution into IVDs using a rat caudal NP punch model. The treatment suppressed IVD tissue degeneration, expression of inflammatory cytokines and nerve growth factor receptor, and reduced pain. The method is minimally invasive and safe, requires no hospitalization or surgery, and is expected to provide higher pain control and disc regeneration than existing surgical treatments.

## Figures and Tables

**Figure 1 cells-12-00505-f001:**
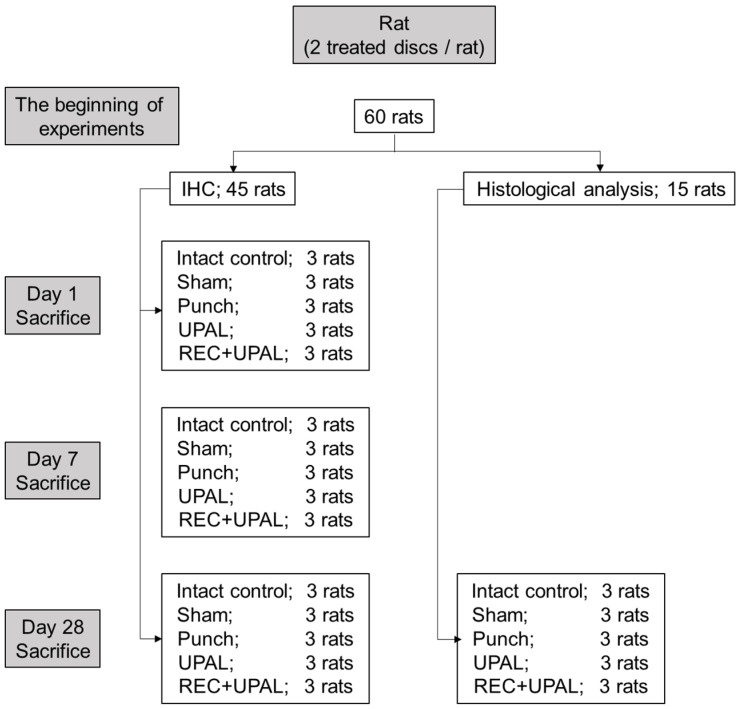
Schematic showing experimental setup, number of rats and treated discs, treatment groups, time points, and analysis.

**Figure 2 cells-12-00505-f002:**
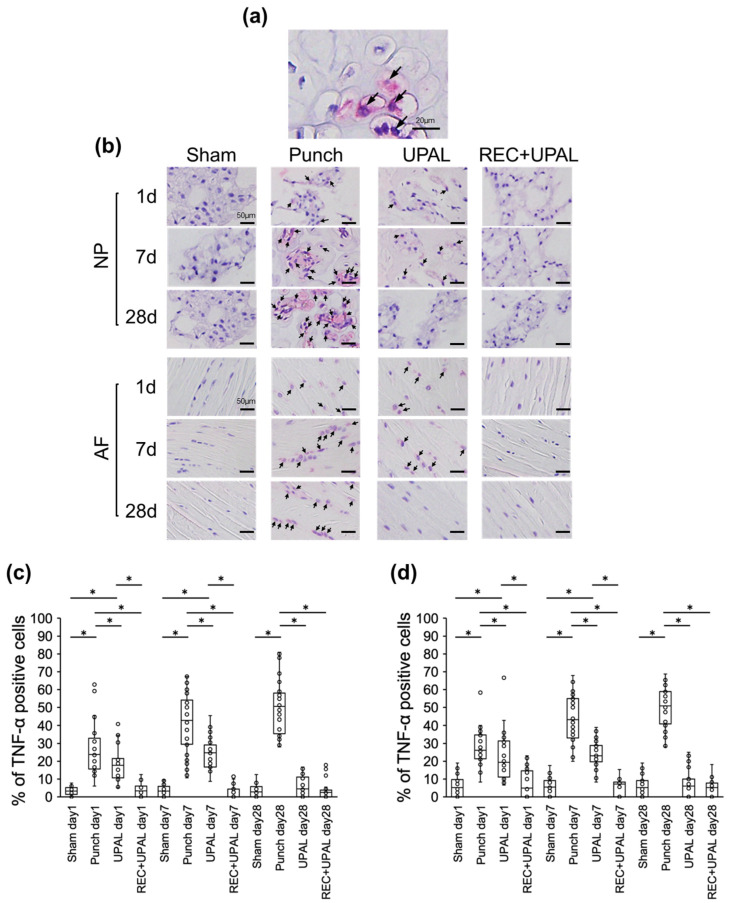
A mixed solution of REC and UPAL inhibits tumor necrosis factor alpha (TNF-α) production. Immunohistochemical analysis was performed on postoperative days 1, 7, and 28 to detect TNF-α levels (**a**–**d**). (**a**) Representative image of TNF-α-positive nucleus pulposus (NP) cells (allows). TNF-α-positive cells are stained in red, and nuclei are stained in purple. Scale bar, 20 µm. (**b**) Representative IHC staining of TNF-α of NP and annulus fibrosus (AF) tissues on postoperative days 1, 7, and 28. Arrows indicate TNF-α-positive cells. Scale bar, 50 µm. Percentage of TNF-α-positive cells of NP tissue (**c**) and AF tissue (**d**). An asterisk (*) indicates a *p*-value less than 0.01. The boxes represent the median and interquartile range, while the vertical lines show the range at each time point (n = 3 rats; n = 6 IVDs in each group).

**Figure 3 cells-12-00505-f003:**
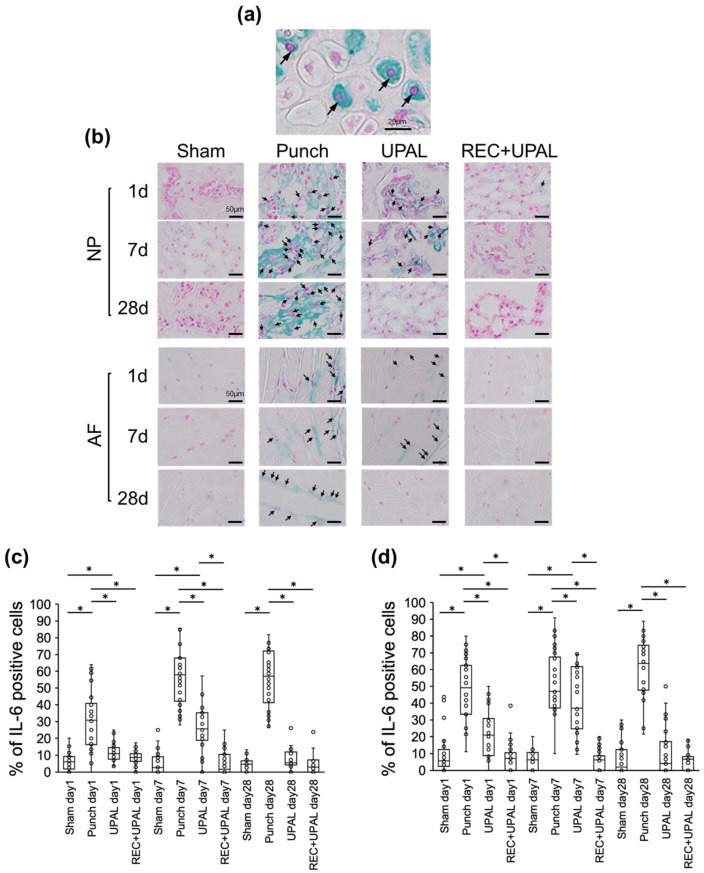
A mixed solution of REC and UPAL inhibits interleukin-6 (IL-6) production. Immunohistochemical analysis was performed on postoperative days 1, 7, and 28 to detect IL-6 levels (**a**–**d**). (**a**) Representative image of IL-6-positive nucleus pulposus (NP) cells (allows). IL-6-positive cells are stained in green, and nuclei are stained in red. Scale bar, 20 µm. (**b**) Representative IHC staining for IL-6 of NP and annulus fibrosus (AF) tissues on postoperative days 1, 7, and 28. Arrows indicate IL-6-positive cells. Scale bar, 50 µm. Percentage of IL-6-positive cells in NP tissue (**c**) and AF tissue (**d**). An asterisk (*) indicates a *p*-value less than 0.01. The boxes represent the median and interquartile range, while the vertical lines show the range at each time point (n = 3 rats; n = 6 IVDs in each group).

**Figure 4 cells-12-00505-f004:**
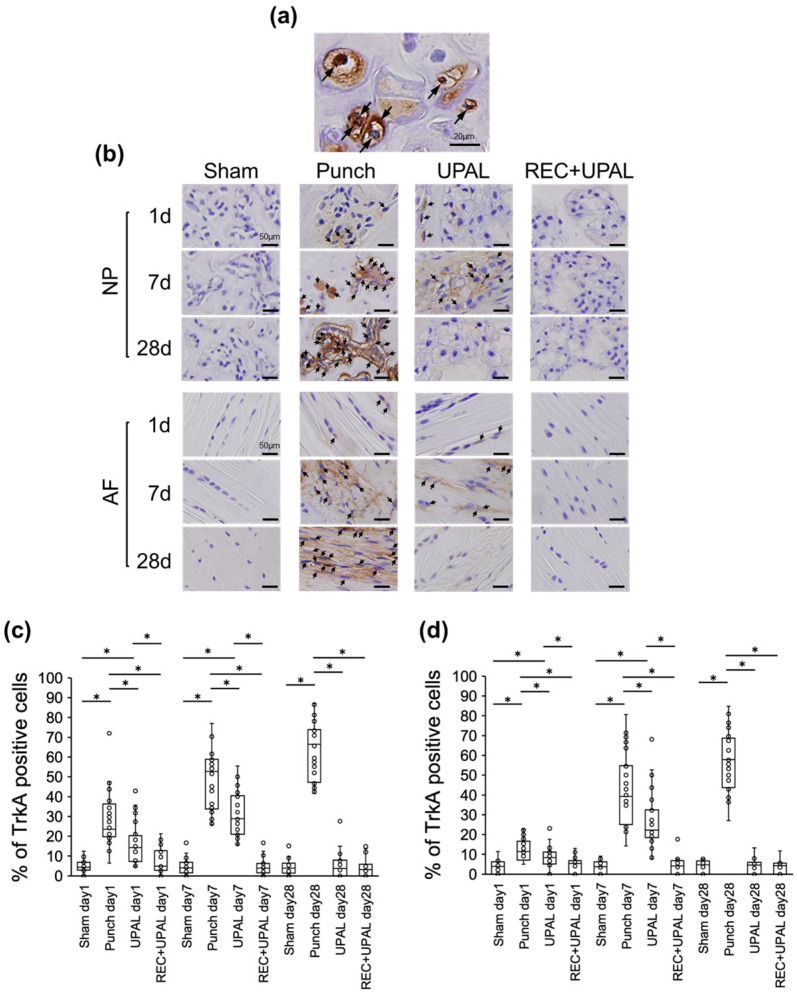
A mixed solution of REC and UPAL inhibits tyrosine kinase A (TrkA) receptor upregulation. Immunohistochemical analysis was performed on postoperative days 1, 7, and 28 to detect TrkA levels (**a**–**d**). (**a**) Representative image of TrkA-positive nucleus pulposus (NP) (allows). TrkA-positive cells are stained in brown, and nuclei are stained in purple. Scale bar, 20 µm. (**b**) Representative IHC staining for TrkA of NP and annulus fibrosus (AF) tissues on postoperative days 1, 7, and 28. Arrows indicate TrkA-positive cells. Scale bar, 50 μm. Percentage of TrkA-positive cells in NP tissue (**c**) and AF tissue (**d**). An asterisk (*) indicates a *p*-value less than 0.01. The boxes represent the median and interquartile range, while the vertical lines show the range at each time point (n = 3 rats; n = 6 IVDs in each group).

**Figure 5 cells-12-00505-f005:**
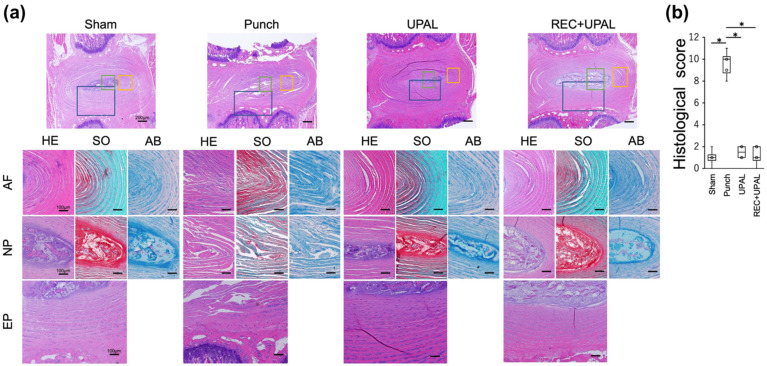
A mixed solution of REC and UPAL prevents intervertebral disc (IVD) degeneration. Histological analysis was performed to evaluate IVD degeneration on postoperative day 28 (**a**,**b**). Mid-sagittal sections (5 μm thick) of rat IVDs were obtained and stained with hematoxylin and eosin (H&E), safranin O (SO), and alcian blue (AB) to evaluate a histological score. (**a**) Representative image of H&E, SO, and AB staining in annulus fibrosus (AF), nucleus pulposus (NP), and endplate (EP) tissues. Scale bar, 200 μm for the top row and 100 μm for all other images. (**b**) Histological degeneration score. An asterisk (*) indicates a *p*-value less than 0.01. The boxes represent the median and the interquartile range, while the vertical lines show the range at each time point (n = 3 rats; n = 6 IVDs in each group).

**Figure 6 cells-12-00505-f006:**
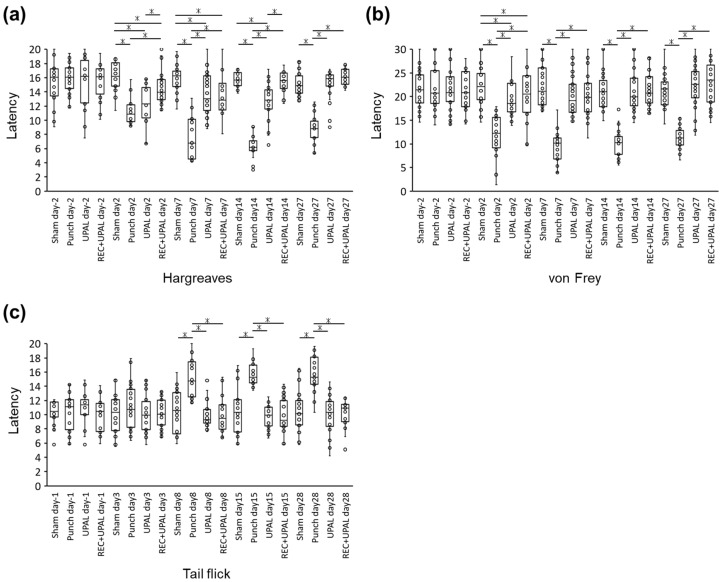
A mixed solution of REC and UPAL reduces nociceptive behavior. The Hargreaves and von Frey tests were performed 2 days before surgery (day—2) and on days 2, 7, 14, and 27 after surgery (**a**,**b**). The tail-flick test was performed 1 day before surgery (day—1) and on days 3, 8, 15, and 28 after surgery (**c**). An asterisk (*) indicates a *p*-value less than 0.01. The boxes represent the median and the interquartile range, while the vertical lines show the range at each time point (n = 6 rats in each group).

## Data Availability

The data that support the findings of this study are available from the corresponding author on reasonable request.

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
