# Peer review of "Injection of Ultra-Purified Stem Cells with Sodium Alginate Reduces Discogenic Pain in a Rat Model"

_cells, 2023, doi:10.3390/cells12030505_

Round 1

Reviewer 1 Report

This study evaluated the effects of injecting a mixture of REC and UPAL non-gelling solution on IVD tissue, inflammatory cytokines, and pain-related behaviors in a rat caudal NP punch model. According to the results of present study, the minimally invasive treatment strategy (performed with a single injection) might be feasible to treat discogenic pain via this regenerative therapy to inhibit IVD degeneration. 

There are still several questions requesting for clarification:

1. Although IHC staining for TNF-α, IL-6 and TrkA of NP and AF tissues are used in this study, the results were indirect evidence. The authors mentioned these inflammatory mediators would be elevated in serum and IVDs of patients with LBP in the Instroduction, the data of blood-based molecular markers might be another strong evidence as outcome measures during a recommended course of REC and UPAL treatments. Because I wonder whether the injection of non-gelling UPAL or REC with UPAL would dilute the IHC staining cells instead of degeneration improving.

2. The same question to the previous REC +UPAL gel in sheep study and this study, why NPC markers, growth factors, and ECM components expression were not measured in the studied disc specimens after euthanasia? Because these results might be the stronger evidence to support the disc regeneration effect by the studied methods.

3. The authors implanted RECs and UPAL in-situ forming gel combination and indicated that UPAL gel prevents cell leakage as a cell carrier and assists in BMSC activation in previous study (eBioMedicine 2022;76: 103845). However, UPAL non-gelling solution was used in this study. How can this design prevent cell leakage?

Reviewer 2 Report

This is nice work, however, a few comments/corrections have to perform prior to acceptance.

1)      This work can absorb more readers if the authors provide a good/comprehensive graphical abstract.

2)      The abstract is lack quantitative results. Only descriptive results are gathered.

3)      The procedure of extracting cells and treating them with sodium alginate did not explain at all. They may refer to the previous publications, however, this has to be explained briefly here.

4)      How the authors measured/quantified the pain in mice to prove the pain has been reduced?

5)      How the authors characterized the stemness status of the bone marrow-derived mesenchymal stem cells?

6)      How to prove that the oncogenes or pro-oncogenes available in mesenchymal stem cells do not activate along with injection in intervertebral disc (IVD)? This becomes more complicated if anyone is going to apply it in humans since HLA in humans is very elaborate compared to the mice.

7)      I did not understand the role of sodium alginate. Is it used as a preservative or did you make a spheroid by using alginate? In this case, you have to optimize the size of the beads and determine the survival rate of the cells either in the beads or free. You could use metabolic screening (or take MRI in two weeks intervals). You could use this paper also:

https://www.mdpi.com/2073-4409/11/3/478

8)      In line 444 mentioned supplementary, but there was no supplementary at the panel.   

Reviewer 3 Report

The authors found that the mixture of REC and UPAL had a better therapeutic efficacy than UPAL alone for the regeneration of IVD. This paper was innovative to some extent, but it still had the followting deficiencies.

1. 3 rats per group is not sufficient to support biological replication, and an increase to 5-6 rats per group is recommended.

2. MRI must be performed to evaluate disc degeneration. If available, mirco-CT can be performed to assess intervertebral height.

3. It is still necessary to conduct relevant experiments to prove the superiority of REC and explore its specific mechanism.

4. There are minor errors in the writing, such as the cell concentration of 1 ×106/ml in line 114 and 121. 

  •  

Round 2

Reviewer 2 Report

The authors answered my question and the quality of the manuscript has been increased.